# Continental drift triggered the Early Permian aridification of North China

Qiang Ren [1,2,3] ✉, Shihong Zhang[2] ✉, Mingcai Hou[1,3] ✉, Dongyu Zheng[1,3], Huaichun Wu [2], Tianshui Yang[2], Haiyan Li [2], Anqing Chen[1,3] & James G. Ogg [1,3,4]

The boundary between wet and arid climate zones in the Tethys Ocean remains challenging to trace, complicating our understanding of global aridification pattern during the Late Carboniferous to Early Permian transition. The North China Block (NCB), situated in the Tethys Ocean, underwent a transition from humid to arid climate during the Early Permian, providing a rare opportunity to trace this climate boundary across this region. Here, we present paleomagnetic evidence indicating that the NCB underwent rapid northward drift between 290 and 281 million years ago. The NCB's movement from a tropical wet to a subtropical arid zone corresponds to a lithological change from coal-bearing to red-bed deposits, demonstrating tectonic drift into a subtropical arid zone as the main driver of aridification in the NCB during this period. This drift also delineates the wet–dry boundary over the Tethys Ocean, consistent with modern climatic zonation patterns.

During the Late Carboniferous to Early Permian transition period, the Pangea supercontinent began to undergo significant aridification[1,2], with most the low-latitude regions transitioning to arid environments[3]. This environmental change led to the collapse of tropical rainforests near-equatorial Pangea[4] and a transition in the deposition of climate-sensitive sediments. For example, indicators of a humid tropical paleoclimate (e.g., coal, laterite, and bauxite) are common in Pennsylvanian strata, but are replaced by indicators of a dry climate (e.g., calcrete and evaporite) in Lower Permian strata. Previous studies have suggested that the large-scale aridification of Pangea's low latitudes was closely related to the migration of the Intertropical Convergence Zone (ITCZ)[2,5]. It is well-established that the ITCZ was unstable over Pangea during the climate transition, with its migration influenced by the deglaciation of Gondwana[1,2]. However, the impact of this instability on the Tethys Ocean remains unclear.

Interestingly, while much of low-latitude Pangea began to experience an arid climate, several microcontinents in the Tethys Ocean continued to experience a tropical rainforest climate[1,3,6]. This marked climatic disparity between low-latitude Pangea and the Tethys Ocean presents challenges in tracking the continuity of climate zones from continental to oceanic regions, resulting in an incomplete understanding of global climate pattern during this climate transition period.

The North China Block (NCB; Fig. 1a), situated within the Tethys Ocean domain, underwent a climatic transition from a tropical rainforest to an arid environment during the Artinskian of the Early Permian[1,7,8]. This transition offers a rare opportunity to delineate the boundary between tropical wet and subtropical arid climatic zones in the Tethys Ocean. However, paleomagnetic data from the NCB during the Early Permian remain sparse, primarily derived from the clastic rocks[9,10], and are constrained by imprecise dating and insufficient inclination shallowing corrections. These limitations have hindered the development of high-resolution paleogeographic reconstructions for this period, further leading to divergent interpretations of the mechanisms driving the NCB's aridification.

Some researchers have proposed that the paleogeographic position of the NCB remained relatively stable during the Early Permian, attributing its aridification to orogenic uplift[11–13] or the exhaust of large

[1]State Key Laboratory of Oil and Gas Reservoir Geology and Exploitation, Chengdu University of Technology, Chengdu, China. [2]State Key Laboratory of Biogeology and Environmental Geology, China University of Geosciences, Beijing, China. [3]Key Laboratory of Deep-time Geography and Environment Reconstruction and Applications, Chengdu University of Technology, Chengdu, China. [4]Department of Earth, Atmospheric, and Planetary Sciences, Purdue University, West Lafayette, IN, USA. ✉e-mail: renqiang@cdut.edu.cn; shzhang@cugb.edu.cn; houmc@cdut.edu.cn

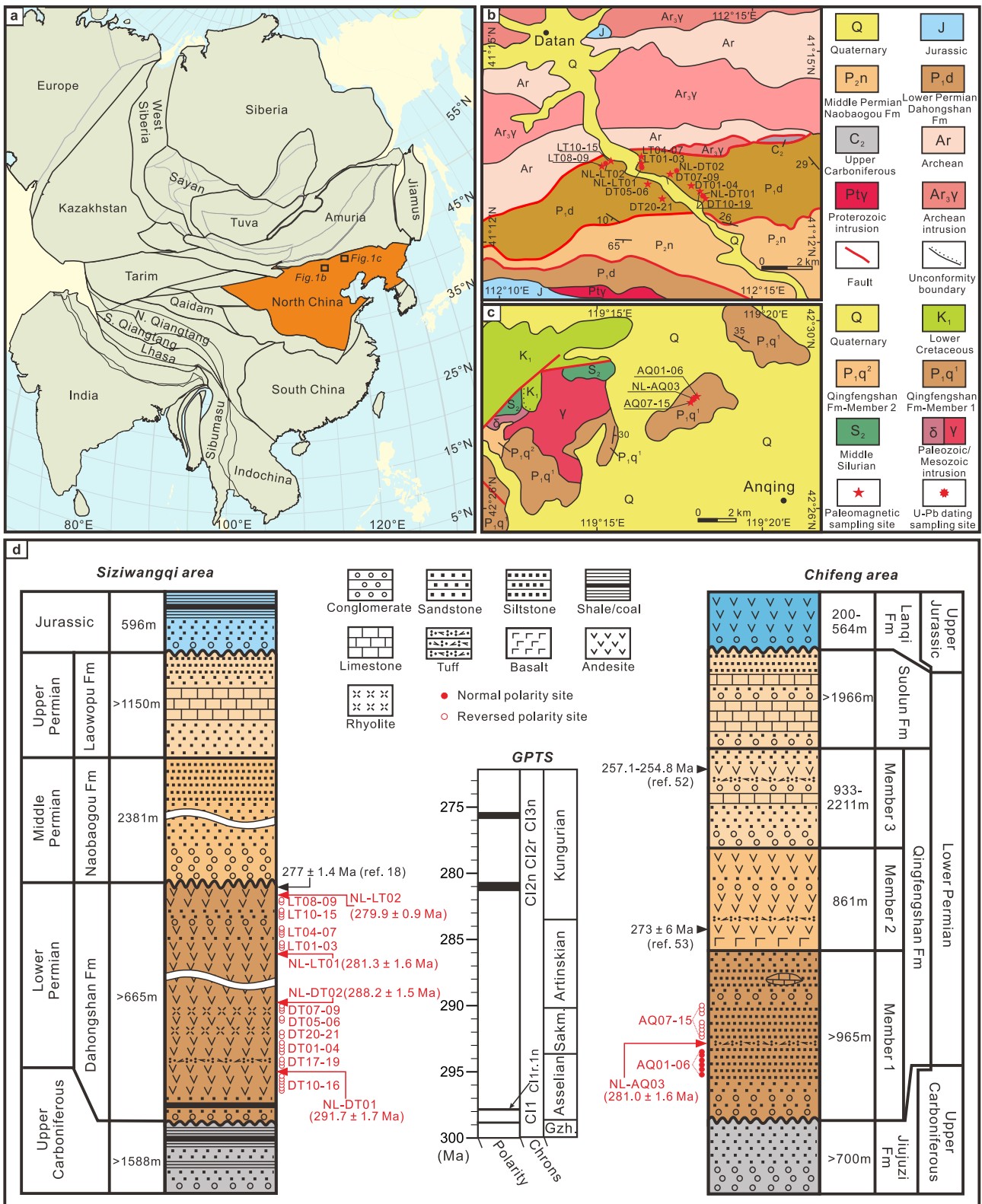

**Fig. 1 | Maps of the sampling localities and strata. a** Tectonic units of eastern Eurasia. Geological maps of sampled sections from the northern NCB: (**b**) the Datan village of south Siziwangqi and (**c**) the Anqing of the northeast Chifeng. **d** Stratigraphic sequence of the study areas. The dating data were cited from Li et al.[19], Ren et al.[54], and Zhang et al.[55]. Geomagnetic Polarity Time Scale (GPTS) is cited from Hounslow and Balabanov[26]. Fm = Formation.

igneous provinces[14,15]. However, these hypotheses cannot explain the spatial heterogeneity in the NCB's climate during this period, characterized by arid conditions in the north and wet conditions in the south. Consequently, greater support has been directed toward the prevailing hypothesis that the NCB's aridification was driven by its northward drift from a tropical wet climate zone to a subtropical arid zone in the Tethys Ocean[6,16,17]. Nevertheless, this hypothesis has yet to be substantiated by reliable Early Permian paleomagnetic data.

In this work, we report two key paleomagnetic data from well-dated (ca. 290 and ca. 281 million years ago; Ma) units in the northern NCB, which reveal a rapid northward motion of the NCB between 290 and 280 Ma. This movement establishes a clear linkage between the NCB's tectonic drift and its transition to arid conditions. The new data provide high-resolution constraints for Early Permian paleogeographic reconstructions, enabling us to trace the boundary between tropical wet and subtropical arid zones in the Tethys Ocean. These findings not only elucidate the mechanisms driving regional climatic changes in the NCB but also shed light on the contrasting climatic patterns between the Pangea supercontinent and the Tethys Oceanic domain.

## Results

Paleomagnetic investigations were carried out in the Siziwangqi (Fig. 1b) and Chifeng (Fig. 1c) areas, near the northern margin of the NCB (Fig. 1a). In Siziwangqi, the Lower Permian Dahongshan Formation (DF; >665 m thick) consists mainly of andesitic and rhyolitic lavas with interbedded volcanoclastic rocks, and is unconformably overlain by molasse facies deposits of the middle Permian Naobaogou Formation and underlain by terrestrial clastic rocks of the upper Carboniferous Shuanmazhuang Formation (Fig. 1d; ref. 18). In the western Yangpanshan Basin, an andesite flow sampled near the top of the DF yields a zircon U–Pb age of $277 \pm 1.4$ Ma[19]. In this study, we collected 183 oriented samples for paleomagnetic analysis comprising andesitic and rhyolitic rocks from 21 sites (DT01–DT21) in the lower member of the DF, along with 155 andesitic and sandstone samples from 15 sites (LT01–LT15) in the upper member, near the village of Datan in southern Siziwangqi (Fig. 1b and Supplementary Fig. 1a). The attitudes of the lava flows were measured from interbedded volcanoclastic rocks (Supplementary Fig. 1b–i). In addition, fresh andesite (NL-DT01 and NL-LT02) and rhyolite (NL-DT02 and NL-LT01) samples collected from the top and bottom of the two paleomagnetic sampling sections were subjected to zircon U–Pb dating (Fig. 1d and Supplementary Fig. 1a). In Chifeng, the volcanic–sedimentary rocks of the lower Permian Qingfengshan Formation (QF) are conformably overlain by sedimentary rocks of the Suolun Formation and unconformably underlain by clastic rocks of the upper Carboniferous Jiujuzi Formation (Fig. 1d; ref. 18). The ~3000 m thick QF is divided into three members[18], of which member 1 (1000-m thick) consists of conglomerates, red siltstones, and tuffaceous interbeds and is the focus of our study. We collected 112 red siltstone samples for paleomagnetic analysis from member 1 at 15 sites (AQ01–AQ21) across an outcrop located ~8 km northwest of the village of Anqing in northeast Chifeng (Fig. 1c and Supplementary Fig. 1j–l). A tuff sample (NL-AQ03) was collected near paleomagnetic sampling site AQ06 for zircon U–Pb geochronology (Fig. 1c and Supplementary Fig. 1m).

### Zircon U–Pb geochronological results

Zircon grains from the andesite (NL-DT01) and rhyolite (NL-DT02) samples from the lower member of the DF are typically 30–80 μm wide and 80–150 μm long, subhedral to euhedral, prismatic, and have clear oscillatory zoning (Supplementary Fig. 2a, d). Their Th/U ratios (0.46–1.30; Supplementary Table 1) suggest a magmatic origin[20]. Thirty-eight analyses of sample NL-DT01 yielded concordant ages (Supplementary Fig. 2b) with a weighted mean $^{206}Pb/^{238}U$ age of $291.7 \pm 1.7$ Ma (MSWD = 1.10; Supplementary Fig. 2c), which is interpreted as the crystallization age. Eighteen analyses of sample NL-DT02 yielded concordant ages (Supplementary Fig. 2e), with a weighted mean $^{206}Pb/^{238}U$ age of $288.2 \pm 1.5$ Ma (MSWD = 0.23; Supplementary Fig. 2f). Therefore, the age of the lower member of the DF and the corresponding paleomagnetic sampling section is 291.7–288.2 Ma (late Sakmarian–early Artinskian).

Zircons from the rhyolite (NL-LT01) and andesite (NL-LT02) samples from the upper member of the DF are typically 30–110 μm wide and 80–150 μm long, subhedral to euhedral, prismatic, and have

clear oscillatory zoning (Supplementary Fig. 2g, j). Their Th/U ratios (0.45–1.42; Supplementary Table 1) indicate a magmatic origin[20]. Twenty-one analyses of sample NL-LT01 yielded concordant ages (Supplementary Fig. 2h), with a weighted mean $^{206}Pb/^{238}U$ age of $281.3 \pm 1.6$ Ma (MSWD = 0.21; Supplementary Fig. 2i), which we interpret as the crystallization age. Twenty analyses of sample NL-LT02 yielded concordant ages (Supplementary Fig. 2k), with a weighted mean $^{206}Pb/^{238}U$ age of $279.9 \pm 0.9$ Ma (MSWD = 0.45; Supplementary Fig. 2l). Therefore, the age of the upper member of the DF and the corresponding paleomagnetic sampling section is 281.3–279.9 Ma (early Kungurian).

Most zircon grains in the tuff sample (NL-AQ03) from the lower member of the QF are colorless and transparent. Their euhedral and prismatic morphology implies they are well preserved. The zircons are typically 20–60 μm wide and 60–120 μm long, and they exhibit highly luminescent oscillatory zoning when subject to cathodoluminescence imaging (Supplementary Fig. 2m). Their Th/U ratios (0.66–2.28) indicate a magmatic origin[20]. Only euhedral, prismatic zircons were chosen for U–Pb analysis. Nine analyses on twelve zircons yielded concordant ages (Supplementary Fig. 2n) and a weighted mean $^{206}Pb/^{238}U$ age of $281.0 \pm 1.6$ Ma (MSWD = 1.08; Supplementary Fig. 2p). Therefore, member 1 of the QF at the paleomagnetic sampling site was deposited during the early Kungurian (ca. 281 Ma).

### Rock magnetic results

Andesite, rhyolite, and sandstone samples from sites DT1-4, DT7, DT10-19, and LT01-15 in the DF were subjected to stepwise thermal demagnetization of the three orthogonal isothermal remanent magnetization (IRM) components[21]. The results show that the soft (0.12 T) and intermediate (0.4 T) IRM components have similar unblocking temperatures of ~580 °C (Supplementary Fig. 3a, c), suggesting that the remanent magnetism in these specimens is carried principally by magnetite.

Stepwise thermal demagnetization of the three orthogonal IRM components[21] in the andesites from sites DT5-6, DT8-9, and DT20-21 in the DF suggests that the IRM intensity is dominated by the soft component (0.12 T). The IRM intensity decreased to zero at ~580 °C (Supplementary Fig. 3b). The hard (2.4 T) and intermediate (0.4 T) components were unblocked at 675 °C (Supplementary Fig. 3b). These results are consistent with the presence of both magnetite and hematite in the andesites.

Stepwise thermal demagnetization of the three orthogonal IRM components[21] in specimens from the red bed (AQ01–AQ21) in the lower member of the QF show that the IRM intensity is dominated by the hard (2.4 T) and intermediate (0.4 T) components. Both IRM intensities decreased rapidly at ~675 °C (Supplementary Fig. 3d), suggesting that hematite is the predominant magnetic carrier in the red beds.

### Paleomagnetic results of the lower member of the DF (290 Ma)

Thermal demagnetization of 155 andesite and rhyolite specimens from 21 sites in the lower member of the DF yields stable magnetic signals (Supplementary Table 2 and Supplementary Fig. 4a–f). The in situ low temperature components (LTCs), mostly removed below 300 °C, are distributed around the local recent geomagnetic field (RGF; Supplementary Fig. 5a) and represent a viscous remanent magnetization (VRM) of the RGF. The stable high-temperature components (HTCs) carried by magnetite and hematite are typically isolated from 500 to 580 °C (site DT1-4, DT7, and DT10-19) and from 560 to 680 °C (site DT5-6, DT8-9, and DT20-21), respectively. This is consistent with the results of rock magnetism experiments (Supplementary Fig. 3a, b).

All site-level HTCs yield uniformly reversed polarity, pointing southeast and upward after tilt correction (Fig. 2a and Supplementary Table 2). All HTCs pass the fold tests of McElhinny[22] and McFadden[23] at the 95% and 99% confidence levels (Supplementary Table 2) and the "k" parameter reaches a maximum at 98.2% unfolding in a stepwise unfolding test (Fig. 2d; ref. 24). The earliest folding of this formation in

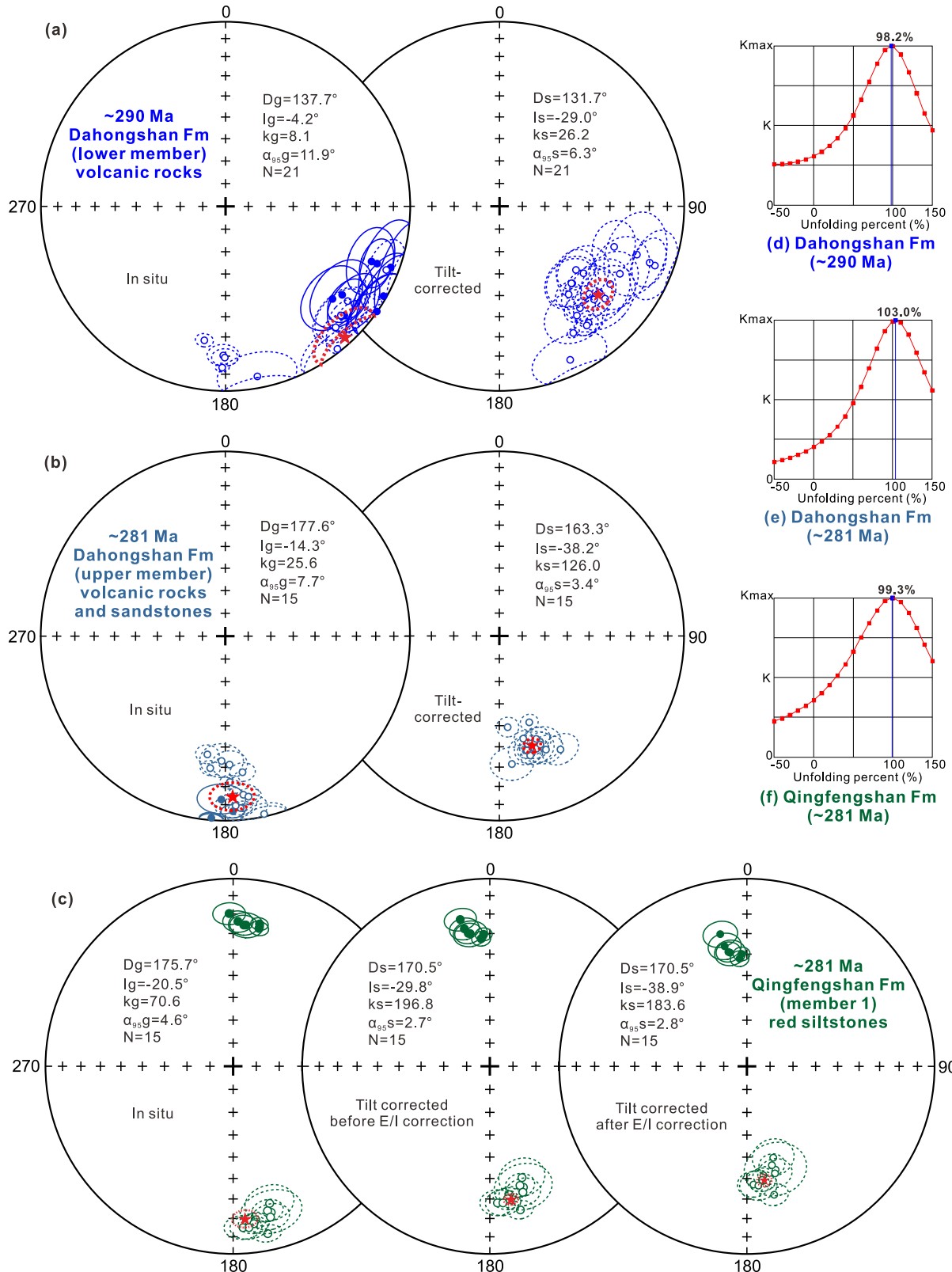

**Fig. 2 | Equal-area stereographic projections of the site-mean HTC directions and fold test. a** Lower member of the Dahongshan Formation (Fm). **b** Upper member of the Dahongshan Fm. **c** Member 1 of the Qingfengshan Fm. Lower (upper) hemisphere directions are represented by solid (open) symbols; red stars indicate the site-level mean directions with 95% confidence limits. **d–f** the stepwise unfolding test. Source data are provided as a Source Data file.

the Siziwangqi region occurred during the latest early Permian–Middle Triassic[18,19], indicating that the HTCs are primary magnetization that was incorporated prior to folding and before the latest early Permian. Nine volcanic breccia specimens from the lower member of the DF

(Supplementary Fig. 7a–c) have stable magnetic signals (Supplementary Fig. 7d). The in situ LTCs, removed below 300 °C, are distributed around the local RGF (Supplementary Fig. 7e) and represent a VRM of the RGF. The stable HTCs decay toward the origin near unblocking

temperatures of 550–580 °C. The HTC directions from the different pebbles are randomly distributed (Supplementary Fig. 7f, g), and those from the same pebble are consistent (e.g., DT22A and DT22B; DT22E and DT22F; Supplementary Fig. 7a, c). A random distribution test gives an R value of 3.39 ($n = 9$), which is significantly lower than the critical R-values of 4.76 and 5.61 at the 95% and 99% confidence levels, respectively[25], indicating a positive conglomerate test. Therefore, these robust fold and conglomerate tests show that the HTCs reflect primary magnetization.

Remanent magnetization was acquired at 291.7–288.2 Ma during the Permian–Carboniferous Reversed Superchron (318–262 Ma; Fig. 1d; ref. 26). Three lines of evidence suggest that paleosecular variation has been time-averaged in the volcanic rocks, as follows: (1) Twenty-one paleomagnetic sites from more than ten different rhyolitic and andesitic lava flows cover a long interval (291.7–288.2 Ma) and are interbedded with many layers of sedimentary rock. (2) The virtual geomagnetic pole (VGP) scatters ($S_B = 13.7°$) for 21 reversed polarity sites are consistent with the PCRS predicted values of the best-fit "Model G" of de Oliveira et al.[27]. (3) The value of $A_{95}$ obtained from the VGPs of 155 lava specimens is 2.9, within an N-dependent $A_{95}$ envelope with a 95% confidence interval (1.6, 3.4), as proposed by Deenen et al.[28,29]. Therefore, given the time-integrated nature of our results, we averaged all of the site-level VGPs determined in the ca. 290 Ma lower member of the DF and obtained a pole at 41.5°N/5.4°E ($A_{95} = 5.8°$; Supplementary Table 2).

### Paleomagnetic results of the upper member of the DF (281 Ma)
A total of 133 andesite and sandstone specimens from 15 sites in the upper member of the DF yield stable magnetic signals during thermal demagnetization (Supplementary Table 2; Supplementary Fig. 4g–i). The in situ LTCs, removed primarily below 300 °C, are distributed around the local RGF (Supplementary Fig. 5b) and represent a VRM of the RGF. The stable HTCs carried by magnetite and hematite are typically isolated from 500 to 580 °C, consistent with the results of rock magnetism experiments (Supplementary Fig. 3c). All site-level HTCs exhibit uniformly reversed polarity, directed south-southeast and upward after tilt correction (Fig. 2b; Supplementary Table 2). All HTCs pass the fold tests of McElhinny[22] and McFadden[23] at the 95% and 99% confidence levels (Supplementary Table 2), with the "k" parameter reaching a maximum at 103% unfolding in a stepwise unfolding test (Fig. 2e; ref. 24), suggesting that the HTC reflects primary magnetization acquired prior to folding. Eleven paleomagnetic specimens from volcanic breccias in the upper member of the DF (Supplementary Fig. 8a, b) also yield stable magnetic signals (Supplementary Fig. 8c). The in situ LTCs, removed below 300 °C, are distributed around the local RGF (Supplementary Fig. 7d), representing a VRM of the RGF. The stable HTCs decay toward the origin near the unblocking temperature of 550–580 °C. HTC directions from different pebbles show a random distribution (Supplementary Fig. 8e, f), while those from the same pebble are consistent (e.g., LT16D and LT16E; LT16H, LT16I, and LT16J; Supplementary Fig. 8a, b). A random distribution test yields an R value of 2.21 ($n = 11$), lower than the critical R-values of 5.29 and 6.25 at the 95% and 99% confidence levels, respectively[25], indicating a positive conglomerate test. Therefore, these robust fold and conglomerate tests confirm that the HTC reflects primary magnetization. The fact that similar results were obtained from the volcanic rocks and sandstones suggests that they represent a long interval (279.9–281.3 Ma) over which we can average out paleosecular variation. Consequently, we averaged all site-level VGPs determined from the ca. 281 Ma upper member of the DF, resulting in a pole at 66.0°N/332.8°E ($A_{95} = 3.3°$; Supplementary Table 2).

### Paleomagnetic results of member 1 of the QF (281 Ma)
Thermal demagnetization results from the 112 red siltstone specimens collected from 15 sites (AQ01-AQ21) in member 1 of the QF revealed

stable magnetic signals (Supplementary Fig. 4j–p; Supplementary Table 2). The in situ LTCs were largely removed below 250 °C and are clustered around the RGF direction (Supplementary Fig. 5c); these represent a VRM of the RGF. Stable HTCs had unblocking temperatures up to 680 °C and indicate that hematite is the main magnetic carrier, which is consistent with results of rock magnetism experiments (Supplementary Fig. 3d). All HTCs have dual-polarity directions (Fig. 2c and Supplementary Table 2) that pass a reversal test[30] at the 95% confidence level (Class B). The angle between the two averages is $\gamma_o = 3.7 < \gamma_{critical} = 5.5$. The results of our geochronological analyses indicate that remanent magnetizations were acquired at ca. 281 Ma (the early Kungurian stage of the early Cisuralian) during a short period of normal polarity at 282–280 Ma (CI2; Fig. 1d; ref. 26). These data account for the documented dual-polarity at our sampling site. The HTCs also pass fold tests[22,23] at 95% and 99% confidence levels, and the "k" parameter reaches a maximum at 99.3% unfolding in the stepwise unfolding test (Fig. 2f; ref. 24). Thus, these robust tests indicate that the HTC represents pre-folding primary magnetization. For the hematite-bearing red beds, we analyzed the Fisher distribution of the site-level VGPs and applied the widely accepted elongation/inclination (E/I) method of Tauxe and Kent[31] to correct for the 112 sample directions. The f value was 0.7 and inclination was corrected from 29.5° to 37.1°, with the 95% confidence interval between 31.2° and 42.8° (Supplementary Fig. 6). According to the palaeomagnetic criteria suggested by Meert et al.[32], the paleomagnetic inclination (after flattening correction) of the red beds from member 1 of the QF is consistent with that of the coeval volcanic rocks in the upper member of the DF, indicating that the flattening correction is valid. We obtained a pole at 68.1°N/323.6°E ($A_{95} = 2.5°$) by averaging all the site-level VGPs in the ca. 281 Ma member 1 of the QF. Furthermore, since both the red beds in member 1 of the QF and the volcano-sedimentary rocks of the upper member of the DF yield similar paleomagnetic results (with their VGPs passing a significance test; Supplementary Table 2), we obtained a pole for the northern margin of the NCB at ca. 281 Ma of 67.1°N/328.4°E ($A_{95} = 2.1°$) by averaging all 30 site-level VGPs.

## Discussion
Our new early Permian paleomagnetic data indicate that the NCB was located at 12.8° ± 5.8°N at -290 Ma and 21.8° ± 2.1°N at -281 Ma (for a reference point at 42.5°N, 119.5°E; Fig. 3a). We compiled the Permian paleomagnetic data from the blocks (including the North China, South China and North Qiangtang blocks) in the Tethys Ocean domain and evaluated them using the seven criteria of Meert et al.[32], which are presented in Supplementary Table 3 with cutoff of $R \geq 4$ (detailed discussion in Supplementary information). Given uncertainties in both plate latitudes and ages, the simple arithmetic averaging method often disregards these errors. Instead, a Monte Carlo simulation was performed to best estimate of the rate of plate motion incorporating this uncertainty[33]. Therefore, Monte Carlo simulations were used to determine the rate of Permian plate motion (Supplementary Fig. 9), yielding a rapid northward latitudinal motion of the NCB at 10.2 cm/yr, with a 95% confidence range of 6.1–15.3 cm/yr from 290 to 281 Ma (based on the ages of the lower and upper members of the DF; see stage 5 in Supplementary Tables 4, 5 and Supplementary Fig. 9). These estimates are approximately of three to four times higher than the average velocity of present-day continental motion (-3 cm/yr; ref. 34).

We made a new paleogeographic reconstruction for the Tethys Ocean domain at -290 and -281 Ma using GPlates (Fig. 4). We compiled the distribution of lithologic indicators of climate in this framework using the Köppen climatic belts classification scheme[3,35]. Based on the occurrence of coals, bauxites, and laterites (indicative of the tropical wet belt) and evaporites and calcretes (indicative of the subtropical arid belt; refs. 6,35–37), we identified a boundary between the tropical wet belt and the subtropical arid belt at -15–25°N/S within the Tethys Ocean

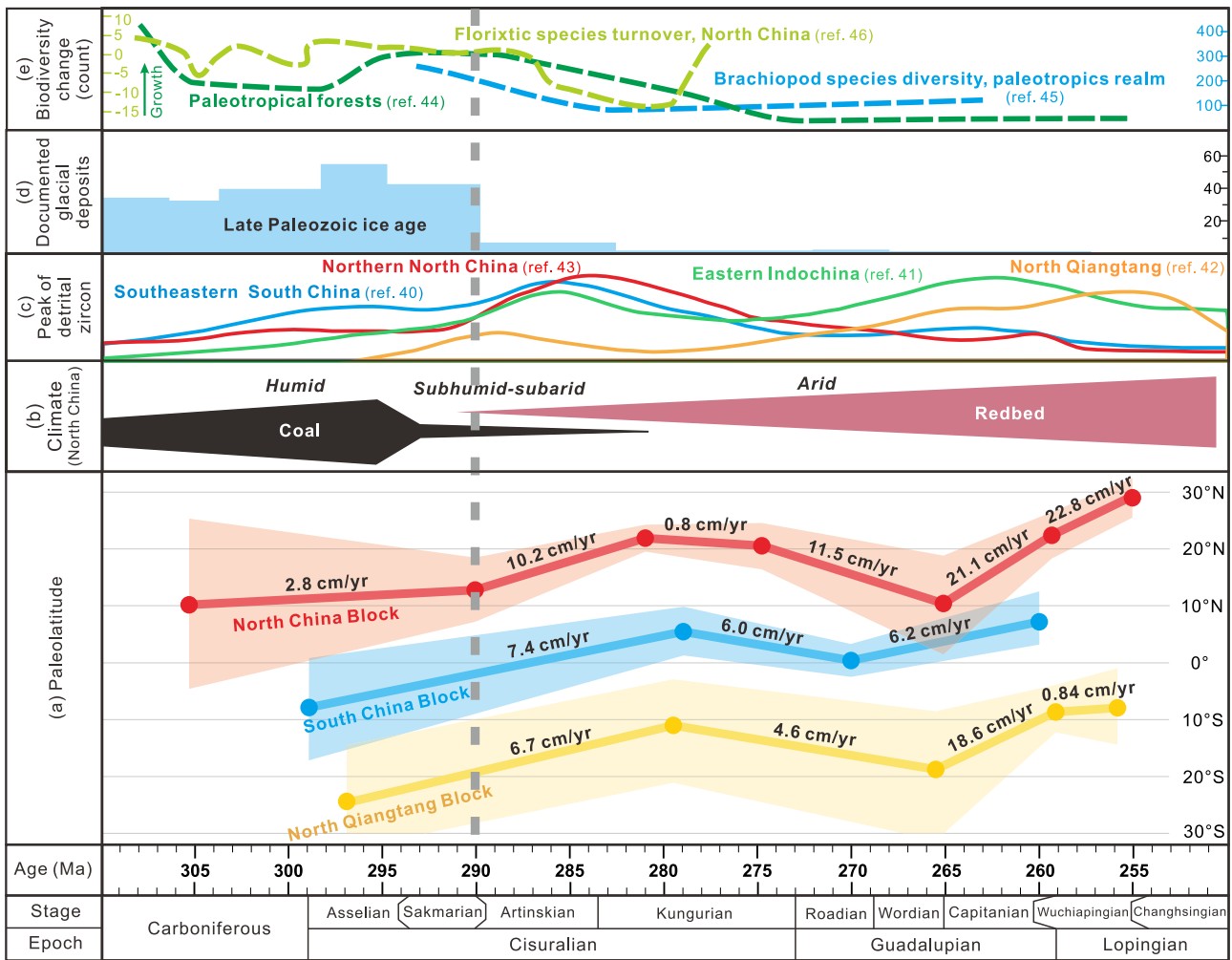

**Fig. 3 | Compilation of the plate latitudinal movement in parallel with climate-biodiversity changes. a** The latitudinal variation versus age for the blocks in Tethys Ocean (paleomagnetic data in Supplementary Table 3). The plate velocity is calculated by Monte Carlo simulation (Supplementary Table 5). Each color range is the paleolatitude error of the different plates. **b** Terrestrial lithologic indicators of climate in the NCB adapted from Wu et al.[8]. **c** Detrital-zircon age spectrum of the Permian strata. **d** Documented glacial deposits adapted from Soreghan et al.[56]. **e** Biodiversity change curves.

domain. This boundary reflects the influence of dual Hadley circulations, where ascending branches near the equator drive precipitation supporting tropical rainforests, and descending branches in the subtropics cause aridification, as evidenced by desert and evaporite deposits[5,6]. Although the Hadley circulation over the Pangea supercontinent was seasonally displaced due to its paleogeographic configuration[1,2], its stability over the Tethys Ocean remains uncertain. Integrating climate-sensitive sediment data, our paleogeographic reconstruction delineates the boundary between the tropical wet belt and the subtropical arid belt across the central NCB at ~280 Ma (~15–25°N; Fig. 4), with a spatial arrangement comparable to modern climatic zonation patterns[3]. This finding suggest that Tethys oceanic regions exhibited greater climatic stability compared to the Pangea supercontinent during the Early Permian.

In the paleogeographic reconstruction, the NCB had not merged into Pangea and was surrounded by ocean during the Early Permian (Fig. 4), undergoing rapid northward movement from 290 Ma to 281 Ma. This movement shifted most of its land area (e.g., the Chifeng, Daqingshan and Ordos areas; Fig. 4) from the tropical wet belt to the subtropical arid belt. Due to the change in the paleoazimuth of the NCB, the western part of the plate had a higher latitude than the eastern part (Fig. 4). This resulted in the western part (including the Alxa and North Qilian areas) being in the arid zone for a long period

during the Permian, leading to the deposition of red beds and explaining the lack of coal seams (Fig. 4). The southern part of the NCB (including the Yuzhou and Huaibei areas; Supplementary Fig. 12) remained in a relatively wet sedimentary environment during the Early–Middle Permian, leading to the deposition of coal seams (Fig. 4). Thus, diachronous deposits were produced in the NCB (Supplementary Fig. 12), changing from wet coal seams to dry red beds. This represents a marked change in sedimentary environment in the NCB, with no evident lithofacies selectivity observed during the transition from coals seams to red beds. This suggests that the transition across the entire block was not influenced by the local lithofacies. Corresponding to the change in rock color, biome diversity played a crucial role in the survival of species in response to climate change[1]. For example, drought-tolerant Ginkgo biloba and conifers began to appear during the early stages of the deposition of the Upper Shihhotse Formation, gradually occupying more competitive ecological niches[6,11,12]. By the Late Permian, the NCB merged into the Pangea tectonic domain as a result of its northward movement. This tectonic integration subjected the entire NCB to arid conditions, influenced by the supercontinental climate system. Therefore, the change from humid to arid climates in the NCB during the Early Permian was mainly the result of its northward drift to subtropical arid latitudes.

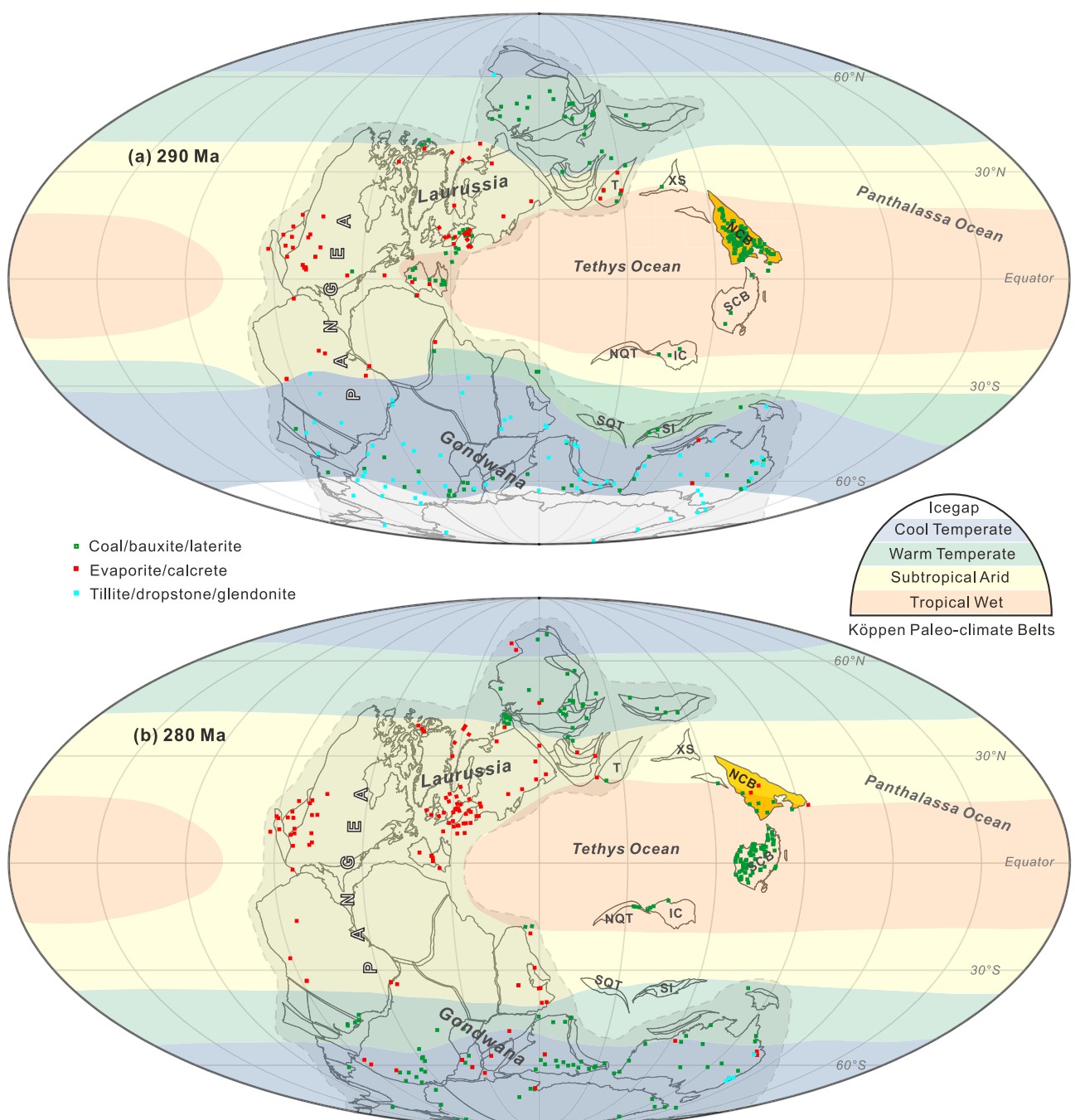

- Coal/bauxite/laterite
- Evaporite/calcrete
- Tillite/dropstone/glendonite

**Icegap / Cool Temperate / Warm Temperate / Subtropical Arid / Tropical Wet**

Köppen Paleo-climate Belts

**Fig. 4 | The paleogeographic reconstructions.** Reconstructions for: (**a**) 290 Ma and (**b**) 280 Ma. The blocks in Tethys Ocean and Pangea are mostly placed according to paleomagnetic constraints (Supplementary Table 3; ref. 57). The zonal Köppen climate belts were reconstructed from lithologic indicators of climate[35], and source data are provided as a Source Data file. NC North China, SC South China, Q Qaidam, T Tarim, IC Indochina, XS Xilinhot-Songliao, SI Sibumasu, NQT North Qiangtang, SQT South Qiangtang.

The rapid latitudinal drift of the NCB not only changed the regional climate, but contributed to global warming. Monte Carlo simulations of the velocity of other Tethys oceanic microcontinents (including the South China and North Qiangtang blocks) suggest similarly rapid northward motion of ~7 cm/yr from ca. 290 to 280 Ma (Fig. 3a; Supplementary Table 5; Supplementary Figs. 10, 11). This rapid latitudinal drift was probably not associated with true polar wander, as Pangea experienced no significant change in latitude between 290 and 280 Ma[38], but instead was due to the tectonic movement of lithospheric plates. The relatively stationary position of Pangea also suggests that its aridification was not driven by tectonic drift, but may

have been influenced by global warming related to increased $\rho CO_2$[1,2,39]. However, in this context of global warming at 290-280 Ma, the terrestrial climate of the Tethys Ocean was highly dependent on the climatic zones these plates traversed. Although these blocks moved rapidly northward, only the NCB traversed multiple climate zones, resulting in gradual intra-plate aridification from north to south. In contrast, the other blocks remained in a tropical humid environment near the equator (Fig. 4). Furthermore, the decrease in the area of Tethyan landmass in the tropics (Fig. 4) related to the northward drift of the NCB likely reduced silicate weathering, resulting in increased atmospheric $\rho CO_2$ and contributing to global warming during the

Sakmarian–Artinskian (Fig. 3d). The rapid convergence of the low-latitude Tethyan landmasses toward Pangea produced intense volcanic activity (detrital zircon age peaks at 290–280 Ma; Fig. 3c; refs. 40–43). The massive volcanic emissions likely contributed to global warming during the Early Permian. Furthermore, we observed a sudden decrease in biodiversity at low latitudes during the Early Permian (Fig. 3e; refs. 44–46). As biodiversity changes are closely linked to environmental and climatic shifts[44], the combined effects of intense volcanic emissions from the convergence of Tethyan plates and the NCB's rapid passage through various climatic zones likely contributed to sudden global warming, leading to a dramatic decrease in both terrestrial and marine biodiversity in the low latitudes.

In conclusion, two reliable paleomagnetic poles (at ~290 and ~281 Ma) are obtained from the northern margin of the NCB. Integrating these paleomagnetic data with evidence from climate-sensitive sediments, we reconstruct the paleogeographic locations of the NCB at 290 and 280 Ma, identifying the boundary between the Tropical Wet Belt and the Subtropical Arid Belt across its central region. This boundary is consistent with modern climatic zonation patterns and highlights the greater climatic stability of oceanic regions compared to the Pangea supercontinent. Additionally, our paleomagnetic data reveals ~10° of rapid (10.2 cm/yr) northward latitudinal motion of the NCB between 290 and 281 Ma. This moved most of land area of the NCB from a tropical wet zone to a subtropical arid zone, corresponding to a transition in lithology from coal-bearing to red-bed deposits. These findings indicate that the tectonic drift of the NCB into the subtropical arid zone from 290 to 280 Ma was the main driver of regional aridification.

## Methods

### Zircon U–Pb

Zircon grains were separated from whole rock samples using conventional heavy liquid and magnetic techniques at the Institute of the Hebei Regional Geology and Mineral Survey in Langfang, China. More than 500 representative zircon grains were hand-picked under a binocular microscope from each sample. Both unknowns and zircon standards were then mounted in epoxy and polished to expose the internal structures of each grain. The morphologies and internal structures of the zircons were studied using transmitted and reflected light microscopy and cathodoluminescence imaging.

Zircon U–Pb geochronology was conducted using laser ablation–inductively coupled plasma–mass spectrometry (LA–ICP–MS) at the Chinese Academy of Geological Sciences, Beijing, China, for sample NL-DT01, at the Mineral Laser Microprobe Analysis Laboratory of China University of Geosciences, Beijing, China, for samples NL-DT02 and NL-AQ03, and at the Gengxin Geological Service Company Limited, Langfang, Hebei Province, China, for samples NL-LT01 and NL-LT02. Procedures followed those described by Liu et al.[47]. Data reduction was performed using ICPMS DataCal 10.2 (ref. 47) for off-line analyses and ComPbCorr#3–151 (ref. 48) for common Pb correction. Zircon ages with >10% discordance were excluded from the final dataset. Uncertainties on individual analyses are reported with 1σ errors; weighted mean ages are reported at the 95% confidence level. Concordia diagrams were plotted and weighted mean ages were calculated using Isoplot 4.15 (ref. 49).

### Paleomagnetic Laboratory Techniques and Measurements

A total of 400 paleomagnetic samples (6–13 cores per sampling site) were collected using a water-cooled portable driller. Samples were oriented using both a magnetic compass and a sun compass. Because the difference in declination between these two methods was <2°, corrections for local magnetic disturbances can be neglected.

All oriented core samples were cut into 2.2-cm long and 2.5-cm diameter specimens for rock-magnetic and paleomagnetic analyses. The specimens were processed and measured in the Paleomagnetism and Environmental Magnetism Laboratory of the China University of Geosciences, Beijing, China. First, thermal demagnetization of the three-axis IRMs[21] was performed on typical specimens to identify ferromagnetic minerals and select the most appropriate demagnetization approach. Fields of 2.4, 0.4, and 0.12 T were successively applied using the IM10–30 pulse magnetizer along the z, y, and x-axes of the specimens, respectively. Then, stepwise thermal demagnetization to 680 °C was performed and measured using an AGICO JR–6 A spinner magnetometer. Based on the results of rock-magnetism assessments, all specimens were then subjected to stepwise thermal demagnetization using an ASC TD–48 Oven (residual magnetic field <10 nT). All remanent magnetizations were measured using a 2 G 755–4 K cryogenic magnetometer and an AGICO JR-6A spinner magnetometer. All thermal demagnetization and remanence measurements were performed in a mu-metal shielded room with residual fields of less than 200 nT. Remanent magnetization directions of all specimens were determined using principal component analysis[50]. The site-mean directions were calculated using the statistics of Fisher[51]. All paleomagnetic data were analyzed using the computational packages of Enkin[52] and Cogné[53].

### Monte Carlo simulation for the calculation of plate movement rates

Monte Carlo simulations were used to determine the uncertainty in the rates of plate movement between pairs of latitudes. Prior to each Monte Carlo simulation, the rate of plate motion as defined by mean latitudes, and mean ages were calculated using:

$$\text{Raw rate} = \frac{\left(\text{mean latitude}_{young} - \text{mean latitude}_{old}\right)}{\left(\text{mean age}_{young} - \text{mean age}_{old}\right)} \quad (1)$$

Raw data can be found in Supplementary Table 4. Both plate latitudes and their ages have associated errors, so a Monte Carlo simulation was performed to best estimate of the rate of plate motion incorporating this uncertainty. For this approach, a large number of random ages and latitudes were selected from a Gaussian distribution to allow the 95% confidence interval of the estimate to be determined. The procedure used is as described below.

First, each latitude and age entry were resampled 10,000 times to generate new subsets of data (Supplementary Figs. 9–11). For example, as shown in Supplementary Fig. 9a, after the Monte Carlo simulation, the seven entries of the NCB were represented in seven discrete datasets, with each dataset containing 10,000 data points.

Second, the rate of plate motion was calculated for each subset, using:

$$\text{rate} = \frac{\left(\text{latitude}_{young} - \text{latitude}_{old}\right)}{\left(\text{age}_{young} - \text{age}_{old}\right)} \quad (2)$$

where latitude$_{young}$ is the location of the plate at the younger age, and latitude$_{old}$ is the location of the plate at the older age. Northern latitudes are positive and southern latitudes negative. age$_{young}$ and age$_{old}$ are young and old ages (Ma), respectively.

Third, estimation of the rates of plate motion used a Monte Carlo simulation. We calculated the rate of plate motion 10,000 times. For each calculation, one data point from the dataset containing older plate positions and another from the dataset containing younger plate positions were sampled. A total of 10,000 calculations yielded a distribution containing 10,000 assessments of the rate of plate motion. The mean, median (50 percentile), 2.5 percentile, and 97.5 percentile of the distribution were calculated (Supplementary Table 5). In the case that data define a strictly normal distribution, the mean, mode, and median of the data should be similar. However, in cases where data are skewed, the mode and median may be more representative. In this study, the median is used to best represent the rate of plate motion, as

it is not affected by outliers. Rates from degrees per million years converted to centimeters per year using the factor 11.1 (cm/yr)/(°/Myr).

## Data availability

Source data are provided in the Zenodo database. The palaeomagnetic data generated in this study have been deposited in the Zenodo database [https://zenodo.org/records/14183814]. The climate-sensitive sediment data generated in this study have been deposited in the Zenodo database [https://zenodo.org/records/14192870]. All data generated in this study are provided in the Supplementary Information file.

## Code availability

The PaleoMac software used for paleomagnetic analyses is available at https://www.ipgp.fr/~fluteau/. Elongation/inclination (E/I) correction were analyzed using the PmagPy (an open source package for paleomagnetic data analysis) at https://pmagpy.github.io/PmagPy-docs/intro.html. The Monte Carlo simulation code generated in this study have been deposited in the Zenodo [https://zenodo.org/records/13859889].

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

## Acknowledgements

This research was supported by National Natural Science Foundation of China (Grants 42472273, to Q.R.), National Natural Science Foundation of China (Grants 41888101, 42330513, to M.C.H.), and Sichuan Science and Technology Program (2023NSFSC1986, to Q.R.). We are deeply grateful to Prof. Christopher R. Scotese, Prof. Xixi Zhao, Dr. Xiujuan Bao and Dr. Hairuo Fu for their insightful discussions and suggestions.

## Author contributions

Q.R. and S.Z. designed research. Q.R., M.H., H.W., T.Y., H.L. and A.C. performed research. Q.R. H.W., T.Y. and H.L. analyzed and interpreted the palaeomagnetic data. Q.R., M.H., A.C. and J.O. contributed the discussion about the environmental and climatic implications. D.Z. contributed to the coding of Monte Carlo simulations. The manuscript was drafted by Q.R. and S.Z and edited by all authors.

## Competing interests

The authors declare no competing interests.
