## [Peer review File · Nature Communications]

REVIEWER COMMENTS

Reviewer #1 (Remarks to the Author):

Review of manuscript “New paleomagnetic evidence constrained tectonic drift to trigger the Early Permian aridification of North China” by Ren et al. for publication in Nature Communications.

This manuscript presents new paleomagnetic and U-Pb geochronology results from 2 Permian section in North China. The authors then use the obtained age and paleolatitude constraints to establish a paleolatitude sequence and argue for tectonic explanations for the observed climatic shifts in North China compared to Pangea. Furthermore, using a Monte Carlo simulation the authors argue for rapid northward motion that caused the aridification in the area.

While I do generally agree with the authors that the data is of high quality and the conclusions are certainly based on the data, I do find some problems in the presentation and argumentation, which should be addressed before the paper can be published.

I will focus on the paleomagnetic data and implications, because this is what I know best. I am not a particular expert for the Permian climatic evolution, especially the differences between North China and Pangea. After reading this manuscript, I do agree that the topic is important and of great interest. However, it sounds more like the authors strengthen a former hypothesis with new high-quality data. This does not address the complexity in climatic evolution between an island and a neighbouring supercontinent including a more or less confined ocean basin. So I am not sure if the presented story is rightly placed at Nature Communications. Having said that, I do believe that robust data is always good especially for controversial topics like this.

Regarding the general presentation, I would suggest the authors get some native English speaker with knowledge of geology to correct the English language. The text sounds often very strange to me, which makes it at times hard to understand (e.g. l.63: “It provides important a unique...”).

Secondly, I am very sceptical that the fold test of the Qingfengshan Fm is correct. Fig. 2c shows the fold test for the Dahongshan Fm, which has some high scatter before untilting and far less scatter after untilting. There the picture in Fig.2c is very likely. However, the directions of Qingfengshan are very similar, which should not result in such a graph, but a much flatter one. Doing a fold test myself, I got a vey different result with an almost flat line (the maximum is still around 100%).

Maybe I missed something, but the authors should definitely check this. I am not arguing against primary origin, which is also shown by the reversal test, but the presentation should be consistent.

Finally, and most importantly, I do have some problems with the arguments for the high plate velocity. Generally, the arguments by the authors makes sense, and it is very justified to argue for climatic changes related to latitudinal changes. However, I have some points of criticism in the authors argumentation: (1) why do the authors not use data from the same section with maybe the same lithology? In Siziwangqi for example, there is an age of 277 not far above the sampling with similar volcanic rocks, which would be a good target as well? Seeing the latitudinal changes in one section would be more convincing, since both areas are also represented by different lithologies. This leads to my second point (2) the inclination shallowing correction is very well carried out, while 112 samples are sufficient but on the lower end (Tauxe and Kent are proposing to use more than 100), but the uncertainty of this analysis is quite large. This is related to the lack of elongation and to the method itself. In my experience, this technique is very susceptible to outliers, so it is not very stable with this low amount of samples. In any case, the uncertainties of this analysis should be taking into account, which the authors don't do. In the study, if I understand it correctly, the authors use the resulting inclination with the corresponding α_{95} value, which is very small. Only using this values the authors get a high shift in latitudes. Taking the uncertainties into account would only require a very low latitudinal shift between the two results when taking the low end of the EI uncertainty interval. Again, I do believe the data is of very high quality and this is a good story, but I think some more discussion should be presented to take the data seriously.

Reviewer #2 (Remarks to the Author):

This is a clear and interesting study arguing for the primary trigger of the late Paleozoic to early Mesozoic aridification of North China by paleomagnetism, which yields two seemingly robust Early Permian paleomagnetic poles from the northern margin of the North China Block (NCB), indicating a fast-northward drift from the tropical humid to subtropical arid zones during the period between ~290 and ~281 Ma. Although the topic, rapid-northward drift of the NCB triggered the aridification of North China, could be traced back at least more than two or three decades ago (e.g., Wu, HN et al., Chinese J Geophys. 1990; Cope et al., Inter. Geol. Rew. 2005), and the comprehensive review of the published late Paleozoic to early Mesozoic paleomagnetic poles shows the rapid latitudinal movement of the NCB (from ~10°N to >20°N) was occurred around the Permian-Triassic boundary (i.e., ~260-240 Ma, Figure 3 of Huang et al., Earth Sci. Rev. 2018), the new Early Permian robust paleomagnetic results from this study provide more robust evidence to constrain the northward drift history of the NCB, which might be the primary trigger for the aridification of North China during the late Paleozoic to early Triassic times.

Firstly, the first Early Permian paleomagnetic pole (~290 Ma) is isolated from a suit of lava flow with robust field tests supporting its high quality for paleogeographic reconstruction. Whilst another Early Permian pole was identified from a clastic sequence, of course, with some tuff interbeds providing absolute geochronological dating age around ~281 Ma. Both dual polarities and positive

fold test result indicate the characteristic remanent magnetization (ChRM) was very likely the primary remanence acquired during the formation of the member I of the Qingfengshan Formation (~1000-m thick). Although either the ~290 Ma pole from the volcanic rocks or the ~281 Ma pole from the siltstones yield a comparable paleolatitude of around ~13°N-16°N at reference site (42.5°N, 119.5°E) for the NCB during the period of ~290-281 Ma, the authors prefer to use the E/I method to perform an inclination shallowing correction for the clastic rocks and then yields a much higher paleolatitude (~22°N) for the member I of the Qingfengshan Formation. Considering the late Paleozoic volcanic and clastic rocks from the North China, South China and North Qiangtang blocks (Zhao et al., 1990; Huang et al., 2018; Cheng et al., 2013; Supplementary Table 3) yielded compatible paleomagnetic results, which indicating none or no significant inclination shallowing was observed from these clastic rocks in these blocks, I wonder if the ~281 Ma clastic rocks are really suffered from significant inclination shallowing? If so, the main finding of this study (very fast northward drift of the NCB during the Early Permian) should be a missense. For this issue, I suggest the authors to do further studies such as the anisotropy of high-field IRM (hf-AIR, Billardello & Kodama, 2009) to confirm if the suspicion is correct. In addition, noting that Meert et al. (2020) state that "... any poles based on flattening corrections will not meet the R5 criterion unless the inclinations are corroborated by paleomagnetic data from either intercalated volcanic rocks that have a similar R-value or other sedimentary rocks within the same sequence that do not require flattening corrections (Section 3.5, page 10)", I do not think the ~281 Ma clastic flattening-corrected pole (this study) is a robust evidence for the very fast northward drift history of the NCB during the late Paleozoic to early Mesozoic.

Secondary, noting that paleomagnetic calculation of plate drift rate is strongly depend on the age of paleomagnetic pole, the estimated age uncertainty of the selected paleomagnetic poles (Supplementary Table 4) should be a key point for the discussions. I note that the authors give a age error of 1.6 Ma for the ~281 Ma clastic pole; however, this age error is a geochronological dating error for the tuff layer in the middle of the sampling section (Figure 1d) rather than the duration of the sampled siltstone sequence (i.e., the member I of the Qingfengshan Formation), which was conformably covered by the member II and member III of the Qingfengshan Formation with two geochronological ages of ~257 to 254 Ma in the top of the member III and of ~273 Ma in the bottom of the member II (Figure 1d). Considering the three members of the Qingfengshan Formation have comparable thickness of ~1000 m, the member I of the Qingfengshan Formation may have a duration around 10 million years, and if taking this age uncertainty into account, estimation of the northward drift rate of the NCB during the Permian should be quite different. Anyway, for this issue, the authors need to provide details about the age-error in the calculation.

Thirdly, the authors use the distribution of coal seams and red bed deposits to be the climate indicators for the tropical humid and subtropical arid zones of the NCB, respectively, and conclude that the NCB's transition from the tropical humid to subtropical arid zone is diachronous ranging from the Early Permian in the western and northern parts to the Late Permian in the southern margin of the NCB (Figures 3-4 and Supplementary Figure 11). This view might be valid; however, whether a suit of red rocks indicates an arid sedimentary environment also depends on its lithology and sedimentary facies. For example, it was well known that Permian rocks in North China basically began to turn red from the Lower Shihhotse Formation (Cisuralian), while the sandstone actually turned red from the Shiqianfeng/Sunjiagou Formation (Lopingian). The Lower Shihhotse and Upper

Shihhotse (Cisuralian, Wu Q et al., *Geology* 2021) formations of North China basically developed in delta plain environment, where sediments were deposited above the sea level, in distributary channel or flood plain facies. In this sedimentary environment, the Lower Shihhotse and Upper Shihhotse sandstones, developed in distributary channel facies, actually does not show red color, while the flood plain facies mudstone, exposed in the air and experienced long-term oxidation after sedimentation, generally displays red color. In other words, the red mudstone in the flood plain facies probably does not represent a totally arid environment. Instead, the assemblage of red mudstones and yellowish sandstones in delta plain environment might reflect a transitional subhumid to subarid environment. Generally, colors of terrestrial sandstones developed in fluvial facies or distributed channel facies, having not been long-termly exposed to the air after sedimentation, could be more reliable indicator of an arid environment. In North China, it was not until the latest Permian (the upper part of the Lopingian Shiqianfeng/Sunjiagou Formation) that the strata sequence was actually transformed into a fluvial facies red bed. In addition, Wu Q et al. (*Geology* 2021) proposed a tectonism-related sedimentary unconformity of ~20 million years from the late Cisuralian to Guadalupian (~280–260 Ma) between the Upper Shihhotse and Sunjiagou formations, based on high resolution CA-ID-TIMS U-Pb dating. Regardless of whether the unconformity is of general significance in North China, they clearly pointed out that the Upper and Lower Shihhotse formations in central North China belong to the humid to subhumid zone, and only evolved into an arid climate during the formation of the Lopingian Sunjiagou Formation. Thus, the transition from tropical humid to subtropical arid environments was more likely a step-by-step process.

Finally, even if the timing of the rapid northward migration of the NCB (from tropical humid to subtropical arid zone) constrained by paleomagnetic data coincides well with the timing of the aridification of the most of North China, whether the rapid migration of North China from tropical humid to subtropical arid zone is the triggering mechanism or the primary factor of aridification in North China still needs further studies. As concluded by Wu Q et al. (2021), “similarities in the floral and climate shift histories between Euramerica and North China suggest that aside from the regional tectonic controls and continental movement, extensive volcanism during the Cisuralian may have played a major role in the global warming and aridification in the aftermath of the late Paleozoic ice age (LPIA)”. In other words, current study may provide robust evidence for the rapid northward migration of the NCB from the tropical humid to subtropical arid zones during the late Paleozoic; but its significance and originality to the aridification of North China as well as the global climate change after the LPIA is open to debate.

Response to reviewers

Dear Reviewers,

Thanks very much for taking your time to review this manuscript. We really appreciate all your comments and suggestions. We have carefully revised our manuscript according to your comments and suggestions. The changes are marked in red words in the revised manuscript. The following is our point-by-point response to the comments and suggestions.

Reviewer #1:

Point 1: I will focus on the paleomagnetic data and implications, because this is what I know best. I am not a particular expert for the Permian climatic evolution, especially the differences between North China and Pangea. After reading this manuscript, I do agree that the topic is important and of great interest. However, it sounds more like the authors strengthen a former hypothesis with new high-quality data. This does not address the complexity in climatic evolution between an island and a neighbouring supercontinent including a more or less confined ocean basin. So I am not sure if the presented story is rightly placed at Nature Communications. Having said that, I do believe that robust data is always good especially for controversial topics like this.

Answer: Thanks for the comment. The internal aridity of Pangea is likely influenced by the interaction of several factors, including the Hercynian orogeny, rising atmospheric CO₂ levels, and deglaciation (Tabor & Poulsen, 2008; Montañez & Poulsen, 2013). However, during the global warming of the Early Permian (end of the LPIA), the North China Block (NCB), an island block in the Tethys Ocean, rapidly drifted towards the Pangea continent. This mode of aridification differs markedly from that of the inner Pangea supercontinent. Additionally, the other Tethyan island blocks in low-latitude regions, such as the South China and Indochina blocks, underwent rapid northward motion. However, only the NCB traversed multiple climatic zones, resulting in progressive intra-plate aridification from north to south, while the other blocks maintained a tropical humid environment near the equator (Fig. 4). If North China had not undergone rapid latitudinal drift from the tropics to the arid zone, it likely would have remained in a humid environment, similar to other low-latitude Tethyan island blocks. We therefore conclude that the rapid drift of North China was the key driver of its gradual spatiotemporal aridification. The results provide unique insights into the mechanisms driving regional climate change in the context of global aridification. We believe that our study on regional climate changes driven by plate tectonics offers significant insights into the co-evolution of Earth's surface systems.

To enhance readers' understanding of the complex climatic evolution of the NCB relative to Pangea and the other Tethyan island blocks, we have expanded the discussion in lines 305–311.

Point 2: Regarding the general presentation, I would suggest the authors get some native English speaker with knowledge of geology to correct the English language. The

text sounds often very strange to me, which makes it at times hard to understand (e.g. l.63: “It provides important a unique...”).

Answer: We sincerely apologize for any language shortcomings in the initial submission. We have since consulted with a native English speaker who has expertise in geology to thoroughly review and improve the language throughout the manuscript. We hope that the quality of the language has been significantly enhanced in this revised version.

Point 3: Secondly, I am very sceptical that the fold test of the Qingfengshan Fm is correct. Fig. 2c shows the fold test for the Dahongshan Fm, which has some high scatter before untilting and far less scatter after untilting. There the picture in Fig.2c is very likely. However, the directions of Qingfengshan are very similar, which should not result in such a graph, but a much flatter one. Doing a fold test myself, I got a vey different result with an almost flat line (the maximum is still around 100%).

Maybe I missed something, but the authors should definitely check this. I am not arguing against primary origin, which is also shown by the reversal test, but the presentation should be consistent.

Answer: Thanks for the suggestion. We have re-examined the data carefully. Since folding occurred after inclination shallowing, we opted to use the data prior to the E/I correction. This pre-correction dataset was analyzed using the stepwise unfolding method in the PMGSC software. It is important to note that before running the analysis, the polarity of the data must be unified; otherwise, the software is unable to recognize it. Please refer to the figure below for further clarification.

Point 4: Finally, and most importantly, I do have some problems with the arguments for the high plate velocity. Generally, the arguments by the authors makes sense, and it is very justified to argue for climatic changes related to latitudinal changes. However, I have some points of criticism in the authors argumentation: (1) why do the authors not use data from the same section with maybe the same lithology? In Siziwangqi for example, there is an age of 277 not far above the sampling with similar volcanic rocks, which would be a good target as well? Seeing the latitudinal changes in one section would be more convincing, since both areas are also represented by different lithologies. This leads to my second point (2) the inclination shallowing correction is very well

carried out, while 112 samples are sufficient but on the lower end (Tauxe and Kent are proposing to use more than 100), but the uncertainty of this analysis is quite large. This is related to the lack of elongation and to the method itself. In my experience, this technique is very susceptible to outliers, so it is not very stable with this low amount of samples. In any case, the uncertainties of this analysis should be taking into account, which the authors don't do. In the study, if I understand it correctly, the authors use the resulting inclination with the corresponding α_{95} value, which is very small. Only using this values the authors get a high shift in latitudes. Taking the uncertainties into account would only require a very low latitudinal shift between the two results when taking the low end of the EI uncertainty interval. Again, I do believe the data is of very high quality and this is a good story, but I think some more discussion should be presented to take the data seriously.

Answer: Thank you for your valuable suggestions. With your important reminder, we were able to collect 155 paleomagnetic samples (andesites and sandstones) and two geochronology samples from the upper member of the Dahongshan Formation (near ~277 Ma section; Li et al., 2018) in the Siziwangqi area in August. We obtained an eruption age of ~281.3–279.9 Ma (Fig. S2), which is consistent with the age of the Qingfengshan Formation.

In terms of paleomagnetic results, we conducted stepwise thermal demagnetization on 133 samples from 15 sites, successfully isolating stable high-temperature components (HTCs). These HTCs passed both fold and conglomerate tests, confirming that they represent primary magnetization. Since the data come from interbedded volcanic and clastic rocks with similar paleomagnetic directions (see Meert et al., 2020), they effectively rule out the influence of paleosecular variation and inclination shallowing.

Additionally, there is no significant difference between the ~281 Ma new paleomagnetic direction and that of the red beds of the Qingfengshan Formation from a similar time period. The virtual geomagnetic poles (VGPs) from both formations passed significance testing (see figure below), further supporting the reliability of the E/I correction for the red beds of the Qingfengshan Formation.

We have included a new sketch of the Dahongshan Formation section in Siziwangqi and incorporated descriptions of the new results from the upper member of the Dahongshan Formation in the main text.

Collection	xMinimum	xMaximum	yMinimum	yMaximum	zMinimum	zMaximum
24dt-vgp	-0.41	-0.32	0.15	0.22	-0.93	-0.89
24aq-vgp2	-0.33	-0.27	0.18	0.26	-0.94	-0.92

1000 bootstrapped Cartesian coordinates for the collections at 95% confidence. ✓ Match!

Reviewer #2 (Prof. Baochun Huang):

Point 1: Firstly, the first Early Permian paleomagnetic pole (~290 Ma) is isolated from a suit of lava flow with robust field tests supporting its high quality for paleogeographic reconstruction. Whilst another Early Permian pole was identified from a clastic sequence, of course, with some tuff interbeds providing absolute geochronological dating age around ~281 Ma. Both dual polarities and positive fold test result indicate the characteristic remanent magnetization (ChRM) was very likely the primary remanence acquired during the formation of the member I of the Qingfengshan Formation (~1000-m thick). Although either the ~290 Ma pole from the volcanic rocks or the ~281 Ma pole from the siltstones yield a comparable paleolatitude of around ~13°N-16°N at reference site (42.5°N, 119.5°E) for the NCB during the period of ~290-281 Ma, the authors prefer to use the E/I method to perform an inclination shallowing correction for the clastic rocks and then yields a much higher paleolatitude (~22°N) for the member I of the Qingfengshan Formation. Considering the late Paleozoic volcanic and clastic rocks from the North China, South China and North Qiangtang blocks (Zhao et al., 1990; Huang et al., 2018; Cheng et al., 2013; Supplementary Table 3) yielded compatible paleomagnetic results, which indicating none or no significant inclination shallowing was observed from these clastic rocks in these blocks, I wonder if the ~281 Ma clastic rocks are really suffered from significant inclination shallowing? If so, the main finding of this study (very fast northward drift of the NCB during the Early Permian) should be a missense. For this issue, I suggest the authors to do further studies such as the anisotropy of high-field IRM (hf-AIR, Billardello & Kodama, 2009) to confirm if the suspicion is correct. In addition, noting that Meert et al. (2020) state that "... any poles based on flattening corrections will not meet the R5 criterion unless the inclinations are corroborated by paleomagnetic data from either intercalated volcanic rocks that have a similar R-value or other sedimentary rocks within the same sequence that do not require flattening corrections (Section 3.5, page 10)", I do not think the ~281 Ma clastic flattening-corrected pole (this study) is a robust evidence for the very fast northward drift history of the NCB during the late Paleozoic to early Mesozoic.

Answer: Thanks for your thoughtful comment. We agree with your observation that the most robustly studied paleomagnetic sections are those containing interlayers of volcanic and clastic rocks. Following the reviewers' valuable suggestions, we were able to collect 155 paleomagnetic samples (andesites and intercalated sandstones) and two geochronology samples from the upper member of the Dahongshan Formation (near ~277 Ma section; Li et al., 2018) in the Siziwangqi area this summer. The obtained eruption age of ~281.3–279.9 Ma (Fig. S2) is consistent with the age of the Qingfengshan Formation.

Regarding paleomagnetism, we successfully isolated stable high-temperature components (HTCs) from 133 samples across 15 sites using stepwise thermal demagnetization. The HTCs passed both the fold and conglomerate tests, confirming that they represent primary magnetization. Given that these data originate from interbedded volcanic and clastic rocks with similar paleomagnetic directions (see Meert et al., 2020), they effectively rule out the influence of paleosecular variation and inclination shallowing.

Moreover, there is no significant difference between the newly obtained directions and those from the red beds of the Qingfengshan Formation of a similar age (~281 Ma). The virtual geomagnetic poles (VGPs) for both formations passed a significance test (see figure below), further confirming the reliability of the E/I correction applied to the red beds of the Qingfengshan Formation.

We have also included a new sketch section of the Dahongshan Formation in Siziwangqi, and the description of the new results from the upper member of the Dahongshan Formation has been incorporated into the main text.

Collection	xMinimum	xMaximum	yMinimum	yMaximum	zMinimum	zMaximum
24dt-vgp	-0.41	-0.32	0.15	0.22	-0.93	-0.89
24aq-vgp2	-0.33	-0.27	0.18	0.26	-0.94	-0.92

1000 bootstrapped Cartesian coordinates for the collections at 95% confidence. ✓ Match!

Point 2: Secondary, noting that paleomagnetic calculation of plate drift rate is strongly depend on the age of paleomagnetic pole, the estimated age uncertainty of the selected paleomagnetic poles (Supplementary Table 4) should be a key point for the discussions. I note that the authors give a age error of 1.6 Ma for the ~281 Ma clastic pole; however, this age error is a geochronological dating error for the tuff layer in the middle of the sampling section (Figure 1d) rather than the duration of the sampled siltstone sequence

(i.e., the member I of the Qingfengshan Formation), which was conformably covered by the member II and member III of the Qingfengshan Formation with two geochronological ages of ~257 to 254 Ma in the top of the member III and of ~273 Ma in the bottom of the member II (Figure 1d). Considering the three members of the Qingfengshan Formation have comparable thickness of ~1000 m, the member I of the Qingfengshan Formation may have a duration around 10 million years, and if taking this age uncertainty into account, estimation of the northward drift rate of the NCB during the Permian should be quite different. Anyway, for this issue, the authors need to provide details about the age-error in the calculation.

Answer: We appreciate the valuable suggestion. In the revised manuscript, we have incorporated new paleomagnetic and geochronologic data from the upper member of the Dahongshan Formation. This includes two age determinations from the top and bottom of the paleomagnetic section, allowing for tighter age constraints. Furthermore, this new data aligns well with the paleomagnetic and chronological data from the red beds of member I of the Qingfengshan Formation. As a result, we are confident in using the new data from the upper member of the Dahongshan Formation for the next phase of Monte Carlo simulations and plate motion rate discussions. Importantly, our conclusion regarding the rapid latitudinal movement of the plate remains unchanged.

Point 3: Thirdly, the authors use the distribution of coal seams and red bed deposits to be the climate indicators for the tropical humid and subtropical arid zones of the NCB, respectively, and conclude that the NCB's transition from the tropical humid to subtropical arid zone is diachronous ranging from the Early Permian in the western and northern parts to the Late Permian in the southern margin of the NCB (Figures 3-4 and Supplementary Figure 11). This view might be valid; however, whether a suit of red rocks indicates an arid sedimentary environment also depends on its lithology and sedimentary facies. For example, it was well known that Permian rocks in North China basically began to turn red from the Lower Shihhotse Formation (Cisuralian), while the sandstone actually turned red from the Shiqianfeng/Sunjiagou Formation (Lopingian). The Lower Shihhotse and Upper Shihhotse (Cisuralian, Wu Q et al., Geology 2021) formations of North China basically developed in delta plain environment, where sediments were deposited above the sea level, in distributary channel or flood plain facies. In this sedimentary environment, the Lower Shihhotse and Upper Shihhotse sandstones, developed in distributary channel facies, actually does not show red color, while the flood plain facies mudstone, exposed in the air and experienced long-term oxidation after sedimentation, generally displays red color. In other words, the red mudstone in the flood plain facies probably does not represent a totally arid environment. Instead, the assemblage of red mudstones and yellowish sandstones in delta plain environment might reflect a transitional subhumid to subarid environment. Generally, colors of terrestrial sandstones developed in fluvial facies or distributed channel facies, having not been long-termly exposed to the air after sedimentation, could be more reliable indicator of an arid environment. In North China, it was not until the latest Permian (the upper part of the Lopingian Shiqianfeng/Sunjiagou Formation) that the strata sequence was actually transformed into a fluvial facies red bed. In

addition, Wu Q et al. (Geology 2021) proposed a tectonism-related sedimentary unconformity of ~20 million years from the late Cisuralian to Guadalupian (~280–260 Ma) between the Upper Shihhotse and Sunjiagou formations, based on high resolution CA-ID-TIMS U-Pb dating. Regardless of whether the unconformity is of general significance in North China, they clearly pointed out that the Upper and Lower Shihhotse formations in central North China belong to the humid to subhumid zone, and only evolved into an arid climate during the formation of the Lopingian Sunjiagou Formation. Thus, the transition from tropical humid to subtropical arid environments was more likely a step-by-step process.

Answer: Thanks for the comment. We agree that different sedimentary facies may yield rocks of varying colors. Based on regional geological data from North China (Wu et al., 2021; Shen et al., 2022), there is no significant lithofacies selectivity associated with climatic changes during the interval of 295–280 Ma across the region. The Lower Shihhotse and Upper Shihhotse formations are predominantly composed of fluvial sediments (see Supplementary Information in Wu et al., 2021, Geology), indicating that the observed black-red transition is not influenced by regional lithofacies.

The disappearance of the 290–280 Ma coal seams in North China marks the transition from a humid to an arid environment, encompassing a brief semi-humid to semi-arid phase, which is an important period of environmental drought. The coal seams in the lower part of the Upper Shihhotse Formation have vanished, and fluvial red beds have begun to appear. Additionally, drought-tolerant species such as *Ginkgo biloba* and conifers emerged during the early stages of the Upper Shihhotse Formation, gradually occupying more competitive ecological niches (Liu et al., 2015; Shen et al., 2022). These findings collectively indicate a notable increase in aridity in North China during this period, though the degree of aridification was not as pronounced as that observed in the Shiqianfeng Formation. Furthermore, North China may have been influenced by the rain shadow effect resulting from the uplift of the northern mountains, compounded by global warming, which exacerbated overall drying during the Shiqianfeng Formation. To clarify the geological manifestations of aridity in North China, we have added a discussion in lines 273–281 of the revised manuscript.

Point 4: Finally, even if the timing of the rapid northward migration of the NCB (from tropical humid to subtropical arid zone) constrained by paleomagnetic data coincides well with the timing of the aridification of the most of North China, whether the rapid migration of North China from tropical humid to subtropical arid zone is the triggering mechanism or the primary factor of aridification in North China still needs further studies. As concluded by Wu Q et al. (2021), “similarities in the floral and climate shift histories between Euramerica and North China suggest that aside from the regional tectonic controls and continental movement, extensive volcanism during the Cisuralian may have played a major role in the global warming and aridification in the aftermath of the late Paleozoic ice age (LPIA)”. In other words, current study may provide robust evidence for the rapid northward migration of the NCB from the tropical humid to subtropical arid zones during the late Paleozoic; but its significance and originality to

the aridification of North China as well as the global climate change after the LPIA is open to debate.

Answer: Good comment. We agree that the background of global warming at the end of the LPIA may have influenced the arid climate in North China, although quantifying this impact is challenging. During this period of global warming, several Tethyan island blocks in low latitudes, such as the South China and Indochina blocks, did not transition to arid environments (Fig. 4). If North China had not undergone rapid latitudinal drift from the tropics to the arid zone, it likely would have remained a wet environment, similar to these low-latitude Tethyan island blocks. Therefore, we conclude that the rapid drift of the North China Block serves as the primary mechanism triggering the gradual spatiotemporal aridity. The subsequent severe aridity in North China may be attributed to both global warming and the rain shadow effect resulting from the uplift of the high mountains along the northern margin. Of course, we agree that the contribution of these factors to the Permian aridity of North China is still worthy of further study. To better articulate the relationship between North China and global climate change, we have added a discussion in lines 305–311 of the revised manuscript.

We would like to express our sincere gratitude to the reviewers for your constructive comments and suggestions. We have made every effort to revise the manuscript in accordance with your recommendations and to enhance its quality to meet the journal's publication standards.

REVIEWERS' COMMENTS

Reviewer #1 (Remarks to the Author):

The authors did a great job in improving the paper at the data side. All my concerns were addressed and I think the high quality dataset is in an even better shape now, leaving no room of speculation that the conclusions of the authors are supported by the data.

I still do think the paper needs some improvement in making the global significance better understandable. The abstract was barely changed, and it concludes that “the tectonic drift of the NCB into a subtropical arid zone between 290 and 280 Ma was the main driver of regional aridification.” So what is the global significance of the results? It sounds like the results support the shifts seen in North China are related to motion of the NCB. That is a nice result, but why should someone who studies e.g. climate evolution in the Permian or climate related to supercontinents or someone else care about these results? I don’t quite follow the argumentation in the discussion: I thought the results fit with models of where the arid belts etc. are, so how can this help to constrain the “climate zone of the Paleo-Tethys”, wasn’t this used as a reason for shifts in sediment type? In line 318 the authors write that “rapid paleogeographic changes in the Tethyan domain may have been the primary reason for the sudden decrease in terrestrial and marine biodiversity”? This has to be further explained! I remember a talk by Celal Sengor (e.g. Paleo-Tethys, Permian extinction, and stratabound copper-sulfide deposits of the Cimmerides, Arizona Geological Digest 22, 2008) that the closure of the Paleotethys caused the Permian mass extinction, is this something the authors want to imply? I am no expert there, so I don’t know the recent publications in this topic, but that would be the only reason I could see in making the paleogeographic motions of the NCB globally interesting. I think this is needed to put the unquestionable great paleomagnetic results in a global perspective apart from regional climatic changes due to motion through climate zones.

Reviewer #1 (Remarks on code availability):

The code is available. I think it is always very helpful to have access to such codes.

Reviewer #2 (Remarks to the Author):

The authors seem to have addressed all the issues raised by the reviewers, but some substantive issues (such as: the influence of the error/uncertainties of the paleomagnetic data as well as their age constraints on the rapid change of the paleoposition of the North China Block (NCB) in Permian; whether the rapid change of the paleoposition of North China was the trigger/primary factor of aridification at the end of the Permian in North China; why does the newly obtained ~280 Ma paleomagnetic pole almost overlap with the Triassic paleomagnetic poles from the Ordos Basin of western NCB?) seem to be no better explanation or solution. In conclusion, the paleomagnetic data in this paper looks to be of high quality and reliable. However, the originality and regional/global significance of the scientific questions discussed are debatable.

Response to reviewers

Dear Reviewers,

Thank you very much for taking the time to review our manuscript again. We greatly appreciate all your comments and suggestions. In response, we have carefully revised the manuscript, incorporating your feedback. The changes are marked in red words in the revised manuscript. The following is our point-by-point response to the comments and suggestions.

Reviewer #1:

Point 1: The abstract was barely changed, and it concludes that “the tectonic drift of the NCB into a subtropical arid zone between 290 and 280 Ma was the main driver of regional aridification.” So what is the global significance of the results? It sounds like the results support the shifts seen in North China are related to motion of the NCB. That is a nice result, but why should someone who studies e.g. climate evolution in the Permian or climate related to supercontinents or someone else care about these results?

Answer: Thanks for your insightful comment. Previous studies based on paleogeographic reconstructions have suggested that the widespread aridity of the Pangea supercontinent might be linked to the equatorward migration of arid climatic zones due to shifts in the Intertropical Convergence Zone. However, due to the difficulty in precisely delineating climate zone boundaries, the implications for the Tethys Ocean domain remain unclear. The North China Block (NCB), drifted in the Tethys Ocean domain underwent a climatic transition from a tropical rainforest to an arid environment during the Early Permian. This transition provides offers a unique opportunity trace the boundary between tropical wet and subtropical arid climatic zones in the Tethys Ocean. Our paleomagnetic data reconstruct the paleogeographic positions of the NCB at 290 and 280 Ma. Combined with the distribution of climate-sensitive sediments, we reconstructed the Tethys Oceanic climate zone showing that the boundary between the tropical wet belt and the subtropical arid belt across the central NCB at ~280 Ma (~15–25°N; Fig. 4). This spatial arrangement is comparable to modern climatic zonation patterns. These findings suggest that the Tethys oceanic regions exhibited greater climatic stability compared to the Pangea supercontinent during the Early Permian.

We greatly appreciate your suggestion to place North China within the global climatic zonation framework. This perspective has enhanced the quality of our manuscript. We have incorporated these insights into the abstract, introduction (lines 36-46), and discussion (lines 261–268 and 318–322). To better illustrate the global significance, we replace the 3D projection in Figure 4 with the Mollweide projection.

Point 2: I don't quite follow the argumentation in the discussion: I thought the results fit with models of where the arid belts etc. are, so how can this help to constrain the “climate zone of the Paleo-Tethys”, wasn't this used as a reason for shifts in sediment type?

Answer: Thank you for pointing out the error in this expression. Integrating these paleomagnetic data with evidence from climate-sensitive sediments, we reconstruct the paleogeographic locations of the NCB, identifying the boundary between the Tropical Wet Belt and the Subtropical Arid Belt across its central region. This provides strong evidence for tracing this boundary in the Tethys Ocean. We have modified it in lines 261-268.

Point 3: In line 318 the authors write that “rapid paleogeographic changes in the Tethyan domain may have been the primary reason for the sudden decrease in terrestrial and marine biodiversity”? This has to be further explained! I remember a talk by Celal Sengor (e.g. Paleo-Tethys, Permian extinction, and stratabound copper-sulfide deposits of the Cimmerides, Arizona Geological Digest 22, 2008) that the closure of the Paleotethys caused the Permian mass extinction, is this something the authors want to imply? I am no expert there, so I don't know the recent publications in this topic, but that would be the only reason I could see in making the paleogeographic motions of the NCB globally interesting. I think this is needed to put the unquestionable great paleomagnetic results in a global perspective apart from regional climatic changes due to motion through climate zones.

Answer: Thanks for your suggestions. In revision, we put the NCB in a global perspective. Our high-resolution paleogeographic reconstruction of Early Permian North China combined with climate-sensitive sediments enables us to accurately trace the boundary between wet and arid climate zones over the Tethys Ocean. By comparing the climate zone patterns within the Pangea supercontinent, we can fully understand the global climate zone patterns during the aridification period.

Additionally, we have elaborated on the primary causes of the sudden decrease in terrestrial and marine biodiversity in lines 308–316. First, the northward drift of the NCB reduced the area of Tethyan landmasses in the tropics (Fig. 4), likely decreasing silicate weathering, which in turn elevated atmospheric pCO₂ levels and contributed to global warming during the Sakmarian–Artinskian. Second, the rapid convergence of low-latitude Tethyan landmasses toward Pangea triggered intense volcanic activity, as evidenced by detrital zircon age peaks at 290–280 Ma (Fig. 3c). The resulting massive volcanic emissions likely exacerbated global warming during the Early Permian. Thus, the combined effects of intense volcanic emissions from Tethyan plate convergence and the NCB's rapid passage through various climatic zones likely drove sudden global warming, ultimately causing a dramatic decline in both terrestrial and marine biodiversity.

Reviewer #2 (Prof. Baochun Huang):

Point 1: the influence of the error/uncertainties of the paleomagnetic data as well as their age constraints on the rapid change of the paleoposition of the North China Block (NCB) in Permian.

Answer: In our research, we accounted for errors in both plate latitudes and ages, which are overlooked by simple arithmetic averaging methods, making the calculated plate

motion rates less reliable. To address this, we employed the Monte Carlo simulation method, which has gained wide acceptance in recent studies (e.g., Swanson-Hysell et al., 2019-GSAB; Fu et al., 2022-SA). This approach, based on statistical principles, is currently the best method for estimating plate motion rates while incorporating these uncertainties. This methodological approach is a key feature of our study. We have provided detailed descriptions of the methods in the Methods section at the end of the text and have clarified our intentions in the Discussion section (lines 241–244).

Point 2: whether the rapid change of the paleoposition of North China was the trigger/primary factor of aridification at the end of the Permian in North China

Answer: Thanks for the comment. This study focuses on the rapid northward movement of North China during the Early Permian (290-280 Ma) and its role in triggering aridification during that period. However, the aridification at the end of the Permian (~255 Ma) occurred approximately 30 million years later. The NCB had become part of the Pangea supercontinent at the end of the Permian, and its climate was predominantly influenced by the supercontinental climate system. The mechanisms driving aridification at the end of the Permian are more complex and multifaceted, involving global climatic shifts and tectonic processes. Currently, we lack definitive evidence to confirm whether the earlier rapid paleogeographic changes of the NCB were a trigger or a primary factor contributing to its later aridification.

Point 3: why does the newly obtained ~280 Ma paleomagnetic pole almost overlap with the Triassic paleomagnetic poles from the Ordos Basin of western NCB?

Answer: Thanks for the question. The robust paleomagnetic tests have proved the reliability of our 280 Ma data. The reliability of our 280 Ma paleomagnetic data has been confirmed through robust paleomagnetic tests. Even if it may overlap with the Early Triassic (T1) paleopole, this could suggest unobvious plate movement between these periods. To explore this further, we compared our 280 Ma paleomagnetic pole with the Early Triassic poles (reviewed in Huang et al., 2018, ESR; see Figure below). Our analysis shows that, despite similar latitudes between the 280 Ma data and the Ordos Early Triassic data, there are notable differences in declination. Importantly, the two 280 Ma consistent paleomagnetic data were obtained from two very far apart sections, reinforcing its representativeness of the stable NCB. The declination differences between the 280 Ma data and the Ordos T1 paleomagnetic data reflect a variation in the paleo-azimuth of the entire NCB. Moreover, the T1 paleomagnetic data from Ordos are based on clastic rock samples, which have been shown to exhibit significant inclination shallowing, with a correction factor (f) of approximately 0.6 (Zhou et al., 2018). After applying the E/I correction, the mean pole of these T1 paleomagnetic data (Zhou et al., 2018) diverges significantly from our 280 Ma paleomagnetic pole (see Figure below). We have added the discussion in Supplementary text.

We would like to express our sincere gratitude to the reviewers for your constructive comments and suggestions. We have carefully revised the manuscript in response to your feedback and have made every effort to improve its quality to meet the journal's publication standards.